# Information Fusion with Knowledge Distillation for Fine-grained Remote Sensing Object Detection

Submission Id: 5596

## ABSTRACT

Fine-grained remote sensing object detection aims to locate and identify specific targets with variable scale and orientation from complex background in the high-resolution and wide-swath images, which needs requirement of high precision and real-time processing simultaneously. Although traditional knowledge distillation technology show its effectiveness in model compression and accuracy preservation for natural images, the challenges of heavy background noise and intra-class similarity faced by remote sensing images limits the knowledge quality of teacher model and the learning ability of student model. To address these issues, we propose the Information Fusion with Knowledge Distillation (IFKD) method to enhance student model performance by integrating information from external images, frequency domain, and hyperbolic space. This includes three key modules: 1) External Disturbance Enhancement (EDE), which uses MobileSAM to enrich teachers' knowledge and reduce students' dependency on teachers; 2) Frequency Domain Reconstruction (FDR) to amplify key feature representations and reduce background noise interference by resampling low-frequency information; 3) Hyperbolic Similarity Mask (HSM) to increase intra-class differences, guiding students in analyzing and utilizing teachers' knowledge, and leveraging the exponential capabilities of hyperbolic space for performance improvement. Experimental results verify that the IFKD method significantly enhances performance in fine-grained recognition tasks compared to existing distillation techniques. Specially, 65.8% and 81.4% $Ap_{50}$ have achieved on optical ShipRSImageNet and SAR Aircraft-1.0 with our method, even which is 0.4% and 4.7% higher than the teacher.

## KEYWORDS

Fine-grained object detection, Knowledge distillation, Remote sensing images, Information fusion.

## 1 INTRODUCTION

Currently, fine-grained remote sensing object detection has become an area of interest, aimed at identifying and locating specific targets in a large range of images, such as ships or aircraft, which is essential for monitoring and protecting the marine and air domains.

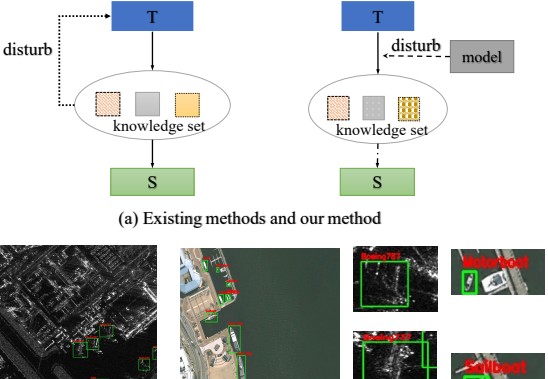

(a) Existing methods and our method

(b) Characteristics of remote sensing images

**Figure 1: Comparison of existing methods and features of remote sensing images. (a) The existing methods generally use the information in the teacher-student system to disturb the information transmission of teachers to students, and our method perturbs the teacher's message by introducing additional information. (b) Complex background of remote sensing data sets and the characteristics of small intra-class differences.**

Although large-scale deep learning models show favorable performance in fine-grained remote sensing detection, it can not meet the real-time processing requirment. Therefore, how to find a better balance between accuracy and efficiency has become a key issue with important research significance. Knowledge Distillation (KD) [12] is one of the effective methods to solve the above problems.

Most existing KD methods [8, 12, 15, 18, 22, 34, 45] typically guide the learning of student models by transferring the prediction results [3], intermediate features [36], and attention maps [22] of the teacher model. To overcome the misleading nature of background information, various decoupling strategies [9, 36] have been proposed, which differentiate between instance-related foreground and instance-unrelated background by generating imitation masks. Through the above method, perturbation of the own knowledge set in the distillation process is shown in Fig. 1 (a). However, these approaches have three limitations. First, they rely too much on imperfect teachers' own knowledge sets. As a result, the potential of the student model cannot be stimulated, and it is difficult to surpass the performance of the teacher model. Secondly, compared with natural images, remote sensing images show complex background and smaller inter-class differences, which leads to a large amount of misleading information in teachers' knowledge set, as shown in Fig

1 (b). Finally, in the fine-grained detection task, the teacher's knowledge set is difficult to be distinguished and analytically learned by the student model due to the small differences within the class.

To address practical challenges, this paper introduces a novel IFKD strategy. This strategy revolves around three key elements: (1) by introducing information from large models, we enrich and perturb the original teacher knowledge set, reducing the student's reliance on the original knowledge set and unlocking its latent potential; (2) suppressing noise in the knowledge set through frequency domain techniques enhances the richness and representational capacity of critical features; (3) the retrieval feature of hyperbolic space is used to help students analyze the class difference problem of knowledge set.

Firstly, by designing the external disturbance enhancement (EDE) module, MobileSAM is introduced to dynamically generate interference features and compete with the teacher model for teaching authority, so as to interfere with the transfer of teacher features and encourage the student model to explore its own potential. Next, through a frequency domain reconstruction (FDR) module, we enhance the key features of the teacher model at the frequency domain level. Typically, low-frequency components carry image style information [33, 40], so by suppressing low-frequency parts and enhancing high-frequency features, we improve the expression power of the teacher model's key features without losing the original feature information. Finally, given the importance of spatial information in remote sensing images, we introduce a hyperbolic similarity mask (HSM) module that maps features to hyperbolic space [13] and constructs masks based on the hyperbolic space distance between teachers and students [28, 38], guiding student to analyze knowledge set information effectively.

We conduct extensive experiments for fine-grained detection tasks on the optical remote sensing dataset ShipRSImageNet [50] and the synthetic aperture radar (SAR) remote sensing dataset SAR-Aircraft-1.0 [51]. Compared with existing advanced knowledge distillation techniques, our IFKD demonstrate significant performance improvements. Furthermore, our method successfully unearthe the potential of the student model, allowing it to surpass the teacher model's performance on multiple evaluation metrics. Our contributions are summarized as follows:

(1) We develop an innovative knowledge distillation method, named IFKD, which integrates external scene information, frequency domain information, and hyperbolic space information for comprehensively improving the performance of the student model.

(2) By designing external disturbance enhancement (EDE) module to compete with teacher model, the original knowledge set is enriched and students' dependence on teachers is weakened. This strategy helps to promote students' pattern discovery and enhance their own feature expression.

(3) Frequency domain reconstruction (FDR) enhances the representation of key features in teacher model and reduces background noise by suppressing low frequency style information.

(4) In order to further improve the overall performance of the student model in processing complex data, we make use of the spatial characteristics of hyperbolic space, pull out the

distribution between classes, and guide students to analyze the teacher's knowledge through the hyperbolic similarity mask (HSM) orientation.

## 2 RELATED WORK

### 2.1 Fine-grained object detection in remote sensing images

In recent years, the accuracy of object detection has significantly improved. Currently, mainstream object detection algorithms based on deep learning can be broadly categorized into one-stage detectors [19, 24, 25, 30] and two-stage detectors [1, 10, 26, 39]. Two-stage detectors first feed features into a Region Proposal Network (RPN) to generate a set of proposals, then utilize these proposals to localize targets, such as in the refinement process of region proposal and localization in Faster R-CNN [26]. While two-stage detectors have higher detection accuracy due to their deep backbone networks, they come with higher computational costs, making them challenging to deploy in practical applications. Conversely, single-stage detectors like RetinaNet [19] offer certain advantages in speed, directly utilizing extracted feature information to predict classification and localization results.

The complexity of remote sensing images poses significant challenges for fine-grained detection tasks [6, 17]. Specifically, remote sensing images exhibit varying aspect ratios, inconsistent orientations, and multiscale transformations [5]. Moreover, fine-grained object detection aims at object-level recognition to further distinguish their subclasses. Therefore, the emphasis lies not on the objects themselves but on mining their discriminative features. For instance, an attribute-guided multi-level enhanced feature network and an influence network for learning additive information are proposed by Zhang et al.[49]. Although image-level recognition methods achieve the purpose of mining discriminative features, they perform poorly in fine-grained recognition of dense objects due to the lack of consideration for the spatial positions of geographic objects, as well as the reliability and stability of multi-object recognition. In contrast, fine-grained object detection tasks require accurate localization of targets to enhance feature recognition and representation capabilities, followed by fine-grained classification of objects. Currently, many outstanding remote sensing detection networks are improved based on Faster R-CNN [19], SSD [20], RetinaNet [19], etc. For example, Li et al. [16] introduce polygonal anchors to replace traditional horizontal anchors and develop an RPN sensitive to rotation, thereby improving the analysis of fine granularity. Wang et al. [35] address the scale issue by proposing a special module to fuse multiscale contextual information. Meanwhile, Xu et al. [41] focus more on the scale variation problem to solve the issue of detection performance degradation and propose a feature alignment detection method. Cheng et al. [4] enhance the separability of different targets through a feature enhancement network. Yang et al. [42] propose the precise R3Det, which improves feature alignment capability through cascade detection.

### 2.2 Knowledge distillation

Knowledge distillation is an effective model compression technique, where the implicit knowledge from a teacher model is transferred to a student model to enhance its performance. Hinton

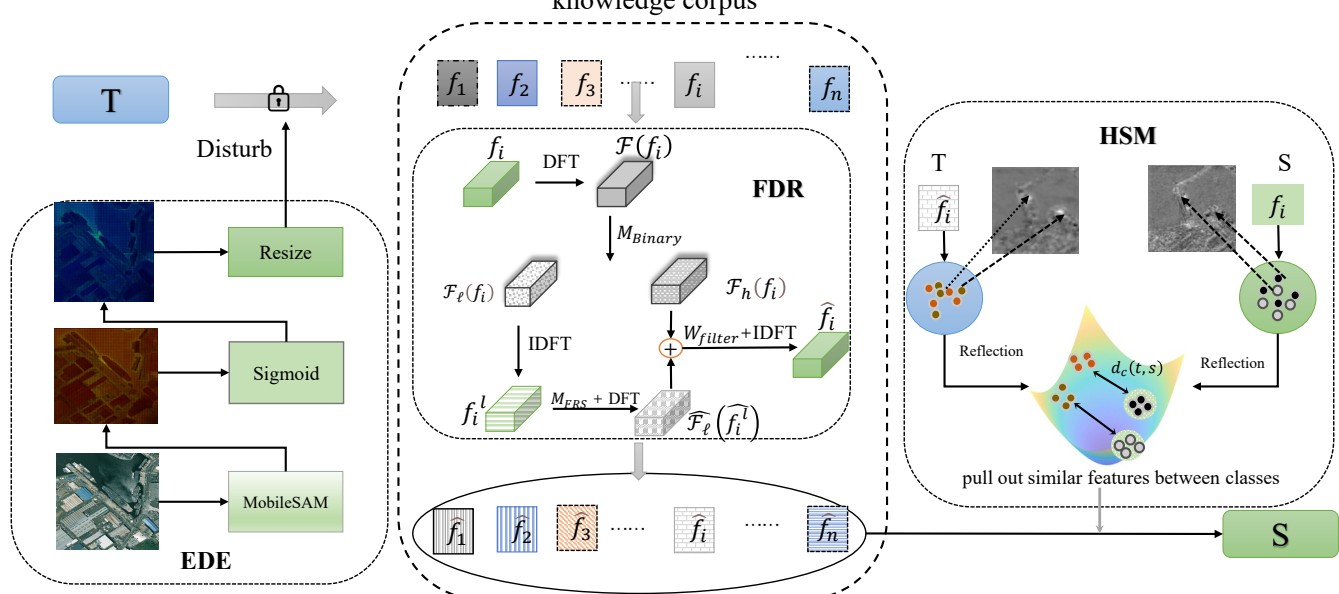

**Figure 2: Overview of our FIKD framework. The teacher network obtains features through feature extractor. The EDE brings up the perturbation information to expand the knowledge base of teachers, and then the characteristics of teachers and students are input to the frequency respectively for feature enhancement, and finally mapped to the hyperbolic space for guided learning.**

[12] pioneers the application of knowledge distillation in classification networks, paving the way for its rapid development. Currently, knowledge distillation is primarily categorized into logit-based methods [12, 48], feature-based methods [11, 27, 45], and relation-based methods [15, 21, 23, 32]. Logit-based methods typically involve transferring knowledge through the final classification layer or logits of the model. Feature-based methods entail transferring knowledge from intermediate features of the teacher model. Relation-based methods focus on exploring relationships between different activations, neurons, or samples.

In recent years, several methods have been proposed to compress object detectors using knowledge distillation. In their work, Chen et al. [2] introduce a methodology that aims to extract student detectors by selectively ignoring the disproportionate emphasis on foreground and background elements. However, the effectiveness of this approach has been found lacking. This shortfall highlights the critical nature of foreground pixels within the context of feature extraction, prompting the proposal of various strategies to decouple these elements. Wang et al. [37] take a nuanced approach by concentrating on the spatial distribution of feature responses, thereby enabling more refined feature extraction. In a similar vein, Ruoyu et al. [29] advocate for the utilization of a two-dimensional mask to diminish background interference, thereby allowing for a focused enhancement of background details. Du et al. [8] propose an innovative strategy to identify salient features external to bounding boxes by categorizing features and mitigating the influence of detrimental features within those boxes. Complementarily, Zhang et al. [47] suggest an attention-guided method for feature

extraction that leverages fine-grained and pixel-level details to approximate mask-based foreground separation. Building upon these ideas, Li et al. [15] explore the establishment of connections among different targets, aiming to bolster both intra-class and inter-class relationships for enhanced discrete learning. Lastly, Yang et al. [43] address the challenge posed by the prevalence of misleading information in remote sensing imagery, underscoring the complexity of accurate image interpretation in this domain. They introduce an adaptive multi-scale feature selection module to guide the student to mimic only core object features. However, whether using these methods to separate foreground and others, they heavily rely on the knowledge set of the teacher model, bringing limitations affecting the distillation performance. On the one hand, the knowledge set of teachers may not be perfect and the quality may not be high enough to stimulate the full potential of students. On the other hand, in fine-grained tasks, the intra-class similarity is high, which makes students unable to analyze different knowledge and learn effectively. Therefore, we propose the IFKD method to improve the knowledge quality by integrating external information to improve the teacher set. Based on the distance of hyperbolic space, the feature distance is amplified to guide students to analyze the knowledge set for efficient learning.

## 3 PROPOSED APPROACH

In the domain of knowledge distillation, the standard framework includes a training dataset $X$, a pre-trained teacher model $T$, and a student model $S$ that is the focal point of training. Conventional approaches typically segregate critical knowledge within the teacher

model and convey it to the student model. Herein, the teacher model $T$ generates a knowledge corpus $K = \{(x, n\tau(x)) | x \in X\}$, wherein each pair $(x, n\tau(x))$ embodies the knowledge to be transferred to the student. Nevertheless, prior methodologies often adhere to a static knowledge set, anchoring the student's learning to this predetermined corpus. This static approach engenders two primary limitations: First of all, students' learning is limited in this unchanging set, and the perfection and quality of the knowledge set limit their learning, thus establishing a clear upper limit for students. Secondly, the quality of knowledge set is affected by the characteristic quality of teachers.

To mitigate these constraints, we introduce an innovative IFKD that amplifies the distillation impact through the assimilation of multifaceted knowledge sources. This method capitalizes on diversifying the information conveyed to the student, enriching their learning landscape beyond the conventional bounds and it also reduces reliance on teachers. As shown in Fig. 2, the teacher network obtains features through feature extractor. then EDE brings up the perturbation information to expand the knowledge base of teachers, and after FRD improves the quality of knowledge set through spectral feature recombination, finally, HSM uses hyperbolic distance between class features to guide learning in direction.

### 3.1 External disturbance enhancement module

Introducing external information to perturb the knowledge provided by the teacher and generate additional knowledge is an innovative approach. In this process, we use MobileSAM to generate additional information pairs for each input image $x$. The generated information pair is spliced with the original $n\tau(x)$ knowledge pair to generate the information knowledge pair $\hat{n}\tau(x)$.

$$\hat{n}\tau(x) = n\tau(x) + sigmoid(MobileSAM(x)).\quad (1)$$

Specially, we do not need to train MobileSAM [46] because it is obtained by distillation of large data sets and large models of SAM [14]. Therefore, compared with the teacher model, the knowledge obtained from each image $x$ contains richer knowledge and can complement the original knowledge set as shown in Fig. 3. Therefore, it is a reasonable practice to introduce MobileSAM to complement the knowledge provided by the teacher. The disturbed knowledge set can be expressed as $K = \{(x, \hat{n}\tau(x)) | x \in X\}$, and the original teacher knowledge set is disturbed, so the teacher's information representation to students is also weakened, and students no longer over-rely on the original teacher. Therefore, the overall function of the EDE module can be described as:

$$\phi_{EDE} = Resize(sigmoid(MobileSAM(x))).\quad (2)$$

Additionally, MobileSAM may have more authoritative knowledge in certain domains compared to the teacher model, which can not only reduce the influence of the teacher on the students but also stimulate the students' own potential. The uniqueness of this approach lies in its ability to help student models acquire knowledge from multiple perspectives, reducing dependence on the teacher model. Moreover, by introducing external information, it can increase the diversity and richness of knowledge, thereby enhancing overall student performance.

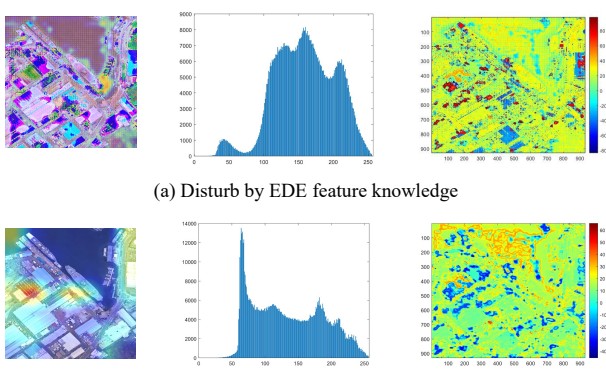

(a) Disturb by EDE feature knowledge

(b) Original feature knowledge

**Figure 3: EDE module disturbance visualization results. (a) Disturb by EDE feature knowledge. (b) Original feature knowledge. MobileSAM generates new knowledge enriched by otherwise low-quality teacher characteristics.**

### 3.2 Frequency domain reconstruction module

In the process of constructing a teaching resource knowledge base, simply adding disturbances and expanding information is not enough. Enhancing the quality of teaching is also crucial. Especially in the field of fine-grained remote sensing target detection, given the scarcity of target information and the complexity of style information in remote sensing images, this study delves into the spectral characteristics of remote sensing images from the perspective of frequency domain analysis. Specifically, high-frequency components of images typically carry more global information, such as the shape of objects, while low-frequency components tend to express more local features [33, 40], such as texture. Based on this, this study employs Discrete Fourier Transform (DFT) to assess the contributions of different frequency domain components to image analysis. Firstly, high-frequency and low-frequency components of image features are extracted. Subsequently, it elaborates on how to utilize frequency domain analysis to enhance the quality of teacher features, the $\phi_{FDR}$ operational steps are as show in algorithm 1. For any given input feature $f(h, w, c) \in R^{H \times W \times C}$, where $H$, $W$, and $C$ represent the height, width, and number of channels, we perform DFT on it to obtain its frequency domain representation:

$$\mathcal{F}(f(h, w, c))_{(u,v,c)} = \sum_{h=0}^{H-1} \sum_{w=0}^{W-1} f(h, w, c) e^{-j2\pi\left(\frac{h}{H}u + \frac{w}{W}v\right)},\quad (3)$$

where $F(f)_{(u,v,c)} \in R^{U \times V \times C}$, $U$, $V$, represent the Frequency domain space range.

We shift the low-frequency components of the image to the center position of the spectrum. Then, we filter these components through a specific window function, where the window $M_{Binary}$ is defined as a binary mask, using mostly random numbers to generate the length of the mask.

$$M_{Binary} = \begin{cases} 1, & \text{, if } \max(|u - \frac{H}{2}|, |v - \frac{W}{2}|) \leq \frac{r \cdot \min(H,W)}{2} \\ 0, & \text{Otherwise} \end{cases},\quad (4)$$

where r is the ratio to control the size of $M_{Binary}$ that distinguishes between high- and low-frequency components of the $f$ gain by:

$$\mathcal{F}_l(f) = M_{Binary} \odot F(f), \tag{5}$$

$$\mathcal{F}h(f) = \left(1 - M_{Binary}\right) \odot F(f), \tag{6}$$

where $\odot$ denotes the Hadamard product. Finally, by performing the inverse Fourier transform, we can convert these frequency domain components back to spatial domain, obtaining the corresponding low-pass features $f^l = \mathcal{F}^{-1}(\mathcal{F}_l(f))$.

Then, by applying a mask operation to the low-frequency components, form $\widehat{f^l} = f^l \odot M_{FRS}$. According to Eq.3, a new low-frequency spectrum $f^l$ and $f^h$, expressed as:

$$f^l = \mathcal{F}^{-1}(\mathcal{F}_l(f)), \tag{7}$$

which is obtained by returning to the frequency domain. $M_{FRS}$ [8] is a global mask that can effectively retrieve important features and eliminate harmful features, so that we interfere with the original low-frequency features. After to further enhance the capture of global structural information, we adopt a learnable filter $W_{filter} \in R^{U \times V \times C}$ aimed at removing information irrelevant to the structure. By performing element-wise multiplication, we obtain the filtered features $\widehat{\mathcal{F}_{filtered}}$ describe as:

$$\widehat{\mathcal{F}_{filtered}} = (\mathcal{F}_h(f) + \widehat{\mathcal{F}_l}(\widehat{f^l}))) \odot W_{filter}, \tag{8}$$

finally, the filtered features are transformed into the spatial domain by Fourier inversion $\widehat{f} = \mathcal{F}^{-1}(\widehat{\mathcal{F}_{filtered}})$, and the optimized features are used to guide students in the learning process of the model.

---

**Algorithm 1** $\phi_{FDR}$ operational steps

---

**Input:** Features in the teacher feature layer $f$

**Output:** The reconstituted features $\widehat{f}$

1: Initially, compute the low-frequency components $\mathcal{F}_l(f)$ and high-frequency components $\mathcal{F}_h(f)$ through the defined Eq.3, Eq.5, and Eq.6.

2: According to formula Eq.7, the time domain semaphore of low frequency $f^l$

3: By applying a mask operation to the low-frequency components, form $\widehat{f^l} = f^l \odot M_{FRS}$

4: According to Eq.3, a new low-frequency spectrum $\widehat{\mathcal{F}_l}(\widehat{f^l})$ is obtained by returning to the frequency domain

5: Adopt a learnable filter $W_{filter} \in R^{H \times W \times C}$

6: Transform the filtered features to the spatial domain $\widehat{\mathcal{F}_{filtered}} = (\mathcal{F}_h(f) + \widehat{\mathcal{F}_l}(\widehat{f^l})) \odot W_{filter}$

7: **return** $\widehat{f} = \mathcal{F}^{-1}(\widehat{\mathcal{F}_{filtered}})$

---

## 3.3 Hyperbolic similarity mask

Extensive research demonstrates that the hyperbolic space effectively embeds linguistic entities, including common words [31], phrases [7], and elements for computer vision tasks [13]. In the field of computer vision, it is acknowledged that hierarchical relationships between images are pervasive. All knowledge methods are

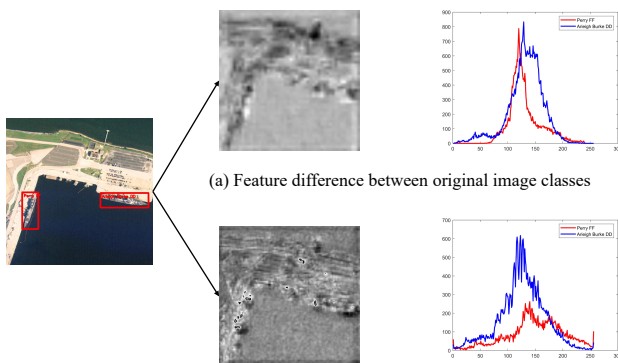

(a) Feature difference between original image classes

(b) Feature difference between hyperbolic space mapping classes

**Figure 4: Visualization of hyperbolic space. (a) Feature difference between original image classes. (b) Feature difference between hyperbolic space mapping classes. Feature maps the hyperbolic space to amplify the intra-class differences.**

based on distance equivalent power law distribution in hyperbolic space, so the similarity is high. Based on this feature, we can map knowledge to hyperbolic networks for more fine-grained learning as shown in Fig. 4.

Therefore, we believe that both teacher and student features in the space have many similar retrievable details. These details can be mapped by the characteristics of the power-law distribution of the hyperbolic space. For teacher features $f_t$ and student features $f_s$ at the same location, the Möbius [13] addition is defined as follows:

$$f_t \oplus_e f_s = \frac{\left(1 + 2c\langle f_t, f_s \rangle + e|f_s|^2\right)f_t + \left(1 - e|f_t|^2\right)f_s}{1 + 2c\langle f_t, f_s \rangle + e^2|f_t|^2|f_s|^2}. \tag{9}$$

Thus, their distance can be expressed as:

$$d_e(f_t, f_s) = \frac{2}{\sqrt{e}} \operatorname{arctanh}(\sqrt{e}| - f_t \oplus_e f_s|), \tag{10}$$

when $e = 1$, the formula simplifies to the geodesic distance, and adjusting $c$ allows the simulation of different Euclidean distances. Consequently, our HSM is formulated as:

$$M_{HSM} = \sum_{h=0}^{H-1} \sum_{w=0}^{W-1} d_e\left(f_t^{(h,w)} - f_s^{(h,w)}\right). \tag{11}$$

Specifically, similar features in the same location have lower weights due to their proximity, while features in the same location but with different features have higher weights. This ensures that, while amplifying similar knowledge, the set of these different weights based on these knowledge becomes a hyperbolic similarity mask (HSM). Through hyperbolic space mask, students are guided to carry out directional learning.

## 3.4 Information fusion distillation loss

Our approach is based on information fusion distillation, which includes feature distillation and detection head distillation. Specifically, feature distillation is given follows:

$$L_{feat} = \sum_{l=1}^{M} \frac{1}{N_l} \sum_{i=1}^{W} \sum_{j=1}^{H} M_{HSM}^{lij} \sum_{c=1}^{C} \left( \phi_{EDE} + \phi_{FDR}(F_{lijc}^{t}) - \phi_{adapt}\left(F_{lijc}^{s}\right) \right)^2, \quad (12)$$

where $M$ denotes the number of layers in the Feature Pyramid Network (FPN), $M_{HSM}$ denotes the hyperbolic similarity mask, $C$ represents the number of channels, $\phi_{EDE}$ disturbs the teacher's knowledge, $\phi_{FDR}$ performs feature recombination, and $\phi_{adapt}$ is a convolutional adaptive function used to adapt $F_s$ to the same dimension as $F_t$. The detection head distillation can be expressed as:

$$L_{head} = \sum_{l=1}^{M} \frac{1}{N_l} \sum_{i=1}^{W} \sum_{j=1}^{H} M_{HSM}^{lij} \sum_{c=1}^{C} \phi_{Binary}\left(y_{lijc}^{s}, y_{lijc}^{t}\right), \quad (13)$$

Here, $l$ represents the $l$-th FPN layer, $y^s$ and $y^t$ are the outputs of the classification head from the teacher model and the student model, respectively, at the $l$-th FPN layer, and $\phi_{Binary}$ is the binary cross-entropy function used in the classification head.

The overall loss is a weighted sum of distillation loss and conventional detector loss:

$$L = L_{GT} + \alpha L_{feat} + \beta L_{head}, \quad (14)$$

where $L_{GT}$ is the training loss of the student detector, and $\alpha$ and $\beta$ are hyperparameters used to balance different distillation losses.

## 4 EXPERIMENTS

### 4.1 Experiment settings

We evaluate two fine-grained remote sensing object detection datasets: SAR-Aircraft-1.0 [51] and ShipRSImageNet [50]. SAR-Aircraft-1.0, released in 2023, is a SAR fine-grained aircraft detection dataset that contains 4,368 images and 16,463 aircraft object instances, categorized into 6 classes: A220, A320/321, A330, ARJ21, Boeing737, Boeing787, and others. Out of these, 3,489 images serve for training and 879 for test. ShipRSImageNet, introduced in 2021, is a challenging public ship detection dataset with 3,426 images and 17,573 ship instances across 50 categories. For this dataset, 2,748 images are used for training and 678 for test. In our experiments, we employ two levels of detectors, namely the two-stage detector Faster R-CNN and the one-stage detector RetinaNet, as baseline detection networks. We choose a deeper network with a ResNet50 feature extractor as the teacher model; meanwhile, lightweight ResNet18 or MobileNetV2 serve as student models. The backbone of all models is initialized with weights pre-trained on ImageNet. All experiments are implemented using the PyTorch and mmdetection frameworks, following 1x learning rate schedule and the default configuration file and train 100 epochs.

### 4.2 Performance on different detection frameworks

In this research, we conducted comprehensive validation of our proposed Inter-frame Feature Knowledge Distillation (IFKD) technique. This validation encompassed both the two-stage Faster R-CNN and the single-stage RetinaNet frameworks. Our experimental findings, as detailed in tables referred to in tables 1 and 2, clearly illustrate that implementing the IFKD strategy considerably enhances

**Table 1: Results using different detection methods in the SAR-Aircraft-1.0 dataset.**

| Method | Faster R-CNN | | | RetinaNet | | |
|---|---|---|---|---|---|---|
| Backbone | mAp | $AP_{50}$ | $AP_{75}$ | mAp | $AP_{50}$ | $AP_{75}$ |
| $T: ResNet50$ | 52.0 | 76.7 | 59.9 | 53.2 | 74.0 | 60.1 |
| $S: ResNet18$ | 49.2 | 76.0 | 57.8 | 50.0 | 73.1 | 57.9 |
| $Ours: ResNet18$ | 52.1 | 77.1 | 60.0 | 53.9 | 75.5 | 60.2 |
| Gain | **+2.9** | **+1.4** | **+2.2** | **+3.9** | **+2.4** | **+2.3** |
| $S: MobileNetV2$ | 55.4 | 80.0 | 64.4 | 53.3 | 76.7 | 59.2 |
| $Ours: MobileNetV2$ | 55.7 | 81.4 | 64.7 | 54.9 | 78.5 | 59.2 |
| Gain | **+0.3** | **+1.4** | **+0.3** | **+1.6** | **+1.8** | **+0** |

**Table 2: Results using different detection methods in the ShipRSImageNet dataset.**

| Method | Faster R-CNN | | | RetinaNet | | |
|---|---|---|---|---|---|---|
| Backbone | mAp | $AP_{50}$ | $AP_{75}$ | mAp | $AP_{50}$ | $AP_{75}$ |
| $T: ResNet50$ | 55.8 | 68.5 | 63.8 | 45.9 | 56.8 | 51.2 |
| $S: ResNet18$ | 48 | 63.2 | 55.2 | 34 | 44.8 | 38.8 |
| $Ours: ResNet18$ | 49.8 | 65.8 | 58.8 | 35.8 | 46.8 | 40.8 |
| Gain | **+1.8** | **+2.6** | **+3.6** | **+1.8** | **+2.0** | **+2.0** |
| $S: MobileNetV2$ | 48.8 | 62.3 | 55.9 | 40.6 | 52.8 | 45.8 |
| $Ours: MobileNetV2$ | 49.4 | 64.4 | 57.4 | 44.4 | 57.2 | 50.1 |
| Gain | **+1.4** | **+2.14** | **+1.5** | **+3.8** | **+4.7** | **+4.3** |

the capabilities of student detectors. Notably, in some tests, the student models even outperformed their corresponding teacher models. Specifically, within the scope of the SAR-Aircraft-1.0 dataset analysis, we observed remarkable outcomes: a student detector employing MobileNetV2 as its feature extractor achieved an $AP_{50}$ of 80.0%, surpassing the teacher model's $AP_{50}$ of 76.7% even without the benefit of knowledge distillation. Upon integrating the IFKD technique, the student model's performance was further amplified to 81.4%. These results not only underscore the significant enhancement potential of the student model when appropriately guided but also suggest that there are further opportunities to push beyond the initial perceived limits of performance.

### 4.3 Comparison with state-of-the-arts

We compare our IFKD with the most advanced knowledge distillation techniques currently available, including FitNet [27], AT [45], FKD [47], InsDist [15], FGD [44], FRS [8], ARSD [43]. Specifically, FitNet and AT represent two methods based on feature distillation.

**Table 3: Comparison with sate-of-the-art methods using different detection, where we use ResNet50 with RetinaNet as the teacher network and MobileNetV2 with RetinaNet as the student network.**

| Model | SAR-Aircraft-1.0 | | | ShipRSImageNet | | |
|---|---|---|---|---|---|---|
| | $mAp$ | $AP_{50}$ | $AP_{75}$ | $mAp$ | $AP_{50}$ | $AP_{75}$ |
| Teacher | 53.2 | 74.0 | 60.1 | 45.9 | 56.8 | 51.2 |
| Student | 53.3 | 76.7 | 59.2 | 40.6 | 52.8 | 45.8 |
| FRS (NIPS) | 52.3 | 75.8 | **60.4** | 45.4 | 58.4 | 51.2 |
| FGD (CVPR) | 52.8 | 75.8 | 59.0 | 44.5 | 57.2 | 50.8 |
| ARSD (TGRS) | 52.6 | 78.0 | 59.7 | 43.8 | 57.8 | 49.7 |
| InsDist (TGRS) | 50.2 | 75.1 | 59.3 | 42.3 | 56.2 | 47.3 |
| FitNet (ICLR) | 52.6 | 78.2 | 55.5 | 41.0 | 53.3 | 45.8 |
| FKD (ICLR) | 52.8 | 77.3 | 59.3 | 40.1 | 54.0 | 44.8 |
| Ours | **54.9** | **78.5** | 59.2 | **46.1** | **59.8** | **51.5** |

FKD employs guided distillation and non-local distillation strategies to address the imbalance between foreground and background pixels and the lack of relationship extraction among different pixels. FRS and FGD emphasize the importance of different features through masking techniques. In recent years, ARSD and InsDist have been designed for knowledge distillation in remote sensing images. To ensure a fair comparison, we implement these methods according to the hyperparameter settings described in their original papers, and test them in the same experimental setup and datasets.

As shown in Table 3, in the SAR-Aircraft-1.0 dataset, although the undistilled student model already surpasses the teacher model in $mAP$ and $AP_{50}$ metrics, other distillation methods rely too much on the knowledge of the teacher model, leading to a decrease in performance after distillation. In contrast, our method introduces additional tacit knowledge, enhancing the knowledge transfer of teacher features and stimulating its potential, resulting in a 2.1% increase in the $mAP$ metric for the student model. On the ShipRSImageNet dataset, considering its inclusion of 50 categories and higher difficulty, the teacher model with deeper feature extractors significantly outperforms the student model. While other distillation methods can improve the performance of the student model, their enhancement is limited by the capability of the teacher model. Conversely, our method, by disturbing the knowledge of the teacher through EDE, weakens its constraints on the student model, allowing the student model's performance to no longer be bound by the upper limit of the teacher's capabilities. Our experimental results demonstrate the effectiveness of this approach, surpassing the performance upper limit of the teacher model on multiple metrics ($mAP$, $AP_{50}$, $AP_{75}$).

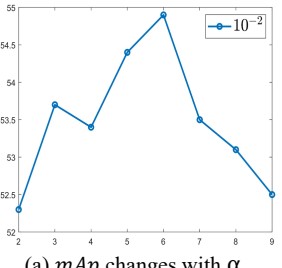 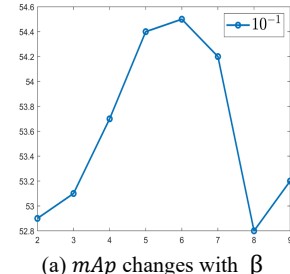

(a) $mAp$ changes with α    (a) $mAp$ changes with β

**Figure 5: Hyperparametric sensitivity studies of $\alpha$ and $\beta$. The teacher model is based on ResNet50 wiht RetinaNet, and the student model is based on MobileNetV2, validated on the SAR-Aircraft-1.0 dataset. (a) The effect of $\alpha$. (b) Impact by $\beta$.**

## 4.4 Ablation study

We conduct ablative experiments on different modules, aiming to delve into the specific influences of each module on the process of knowledge distillation. As shown in Table 4, the experimental results clearly demonstrate that each individual module can effectively enhance the performance of the student model. Specifically, through a detailed comparison based on the $AP_{50}$ metric, we find that the EDE module brings the most significant performance improvement, reaching 4.1% on top of the baseline (AT [45]) module; closely following is the HSM module, achieving a performance growth of 4.0%; while the FDR module enhances by 3.4%. Combining the modules pairwise also yields decent results. It is noteworthy that when these three modules are used in combination, we achieve the highest performance improvement of up to 6.8%. This result indicates that by integrating additional sources of knowledge and leveraging the guiding mechanism in hyperbolic space, we can not only expand and enrich the model's knowledge base, but also effectively stimulate the potential of the student model, thereby significantly enhancing the overall performance of the model.

## 4.5 Sensitivity to hyperparameters

In our IFKD method, by introducing two key hyperparameters, $\alpha$ and $\beta$, we aim to balance different distillation losses and optimize the learning efficiency of the student model. To analyze the impact of these two parameters on model performance in depth, we adopted a strategy of fixing one parameter while dynamically adjusting the other to observe performance changes. Specifically, experiments conducted on the RetinaNet model using the SAR-Aircraft-1.0 dataset's distillation results are presented in Fig. 5.

From Fig. 5(a), it can be observed that when the value of $\alpha$ is too small, the contribution of knowledge distillation to the student model is extremely limited. However, as $\alpha$ gradually increases, we observe a corresponding increase in the $mAP$ performance metric of the student model, indicating that moderate increments in $\alpha$ can effectively facilitate the acquisition of more valuable knowledge from the teacher model by the student model. However, when $\alpha$ increases to a certain extent, the model performance sharply declines, which may be due to the disruptive effect of excessively large

**Table 4: Ablative study of different distillation modules on ShipRSImageNet dataset,where we use ResNet50 with RetinaNet as the teacher network and MobileNetV2 with RetinaNet as the student network**

| AT(baseline) | ✓ | ✓ | ✓ | ✓ | ✓ | ✓ | ✓ | ✓ |
|---|---|---|---|---|---|---|---|---|
| EDE | | ✓ | | | ✓ | ✓ | | ✓ |
| FDR | | | ✓ | | ✓ | | ✓ | ✓ |
| HSM | | | | ✓ | | ✓ | ✓ | ✓ |
| $AP_{50}(\%)$ | 53.0 | 57.1 | 56.4 | 57.0 | 57.4 | 57.7 | 57.6 | **59.8** |

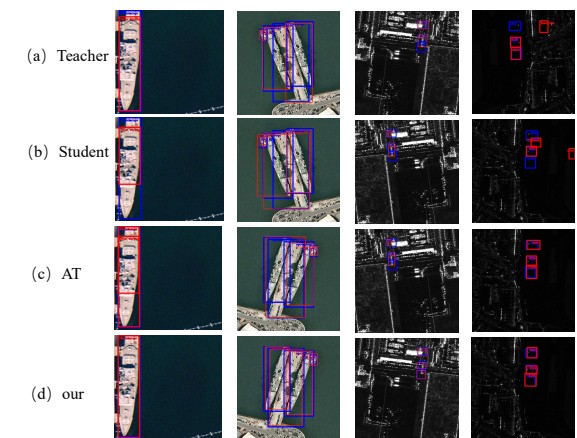

**Figure 6: Qualitative analysis on ShipRSImageNet dataset with distilled and baseline Retianet-ResNet50. The red box represents the detected structure and the blue represents the real result. (a) Teacher result. (b) Student result. (c) AT result. (d) Our result**

distillation losses during training, leading to the student model's inability to learn effectively.

Similarly, Fig. 5(b) shows the trend of $\beta$ parameter's impact, which follows a pattern similar to $\alpha$. A too small $\beta$ limits the learning potential of the student model, as it makes the student model overly reliant on the teacher model, hindering its ability to fully utilize its own learning capacity. However, appropriately increasing $\beta$ can promote the independence and learning efficiency of the student model, yet excessively high $\beta$ values can also have a negative impact on model performance. In this article, we end up choosing $\alpha$ to be 0.06 and $\beta$ to be 0.5.

### 4.6 Visualization

We conduct a qualitative comparison of the results among the teacher model, student model, AT [45] method, and our method. As shown in Fig. 6, when the target merges with the environment, both the teacher (Fig. 6(a)) and the non-distilled student model (Fig. 6(b))

fail to extract fine-grained knowledge, resulting in erroneous detection boxes. The AT method (Fig. 6(b)), due to its excessive reliance on the teacher model, generates multiple detection boxes and fails to surpass its capacity limit, thereby operating with erroneous detections. In contrast, our method (Fig. 6(d)) leverages additional knowledge to assist student learning, thereby reducing its reliance on teacher feature representations. In the SAR-Aircraft-1.0 dataset, as shown in Fig. 6, when the target feature is particularly close to the environment and the inner class similarity is particularly high, the teacher (Fig. 6(a)) model can successfully detect the target due to its excellent feature extraction ability, but it cannot correctly classify the example. The student model (Fig. 6(b)) not only has the wrong detection, but also produces the wrong classification result. Although the student model with knowledge distillation AT [45] (Fig. 6(c)) inherits the excellent performance of teachers, it is limited by over-dependence on the characteristics of teachers, which leads to the wrong category classification. In contrast, our approach (Fig. 6(d)) is able to absorb key characteristics of teachers and identify erroneous knowledge in them, reducing excessive reliance on teacher information, and thus achieving more accurate target identification and positioning. Ultimately, our method guides the student to avoid such errors, indicating that our IFKD possesses better guidance and discrimination capabilities.

## 5 CONCLUSION

This paper proposes an effective distillation method for fine-grained object detection in remote sensing images. We discuss the influence of the teacher feature knowledge base on the student and analyze the adverse effects of excessive reliance on the teacher knowledge base. Based on this finding, we propose a solution based on the IFKD method by incorporating additional information. Specifically, we introduce EDE to weaken the representation of the teacher's own features on the student and enrich the knowledge base to broaden the student's horizon. We refine the knowledge base through FDR to improve the quality of knowledge, and finally guide the student through HSM to stimulate its potential, thereby enhancing the overall performance of the student detector. The IFKD method effectively improves the performance of modern detectors and can be widely applied to both one-stage and two-stage detection frameworks.

Future plans based on this paper could involve further refining and extending the proposed IFKD method for fine-grained object detection in remote sensing images. Conduct research to optimize the EDE technique to effectively weaken the influence of the teacher's features on the student while enriching the student's knowledge base.

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
