# OpenReview forum: "Information Fusion with Knowledge Distillation for Fine-grained Remote Sensing Object Detection"
_acmmm.org/ACMMM/2024/Conference — MM2024 Poster_

### Official Review · Reviewer_Qc3r · 2024-05-11

**Rating:** 3
**Confidence:** 4

**Summary:**

The paper proposes an Information Fusion with Knowledge Distillation (IFKD) method to enhance fine-grained object detection in remote sensing imagery. The method introduces three key modules: External Disturbance Enhancement (EDE), Frequency Domain Reconstruction (FDR), and Hyperbolic Similarity Mask (HSM) to address the challenges of heavy background noise and intra-class similarity. The IFKD method demonstrates significant improvements over existing knowledge distillation techniques and shows the potential for the student model to exceed the teacher model's performance.

**Strengths:**

1. Novel Integration of Techniques: The paper innovatively integrates external image information, frequency domain processing, and hyperbolic space for knowledge distillation.
2. Empirical Results: The method shows strong empirical results, outperforming existing distillation techniques on remote sensing datasets.
3. Student Model Performance: The student model's ability to surpass the teacher model in certain metrics is a notable achievement.

**Limitations:**

1. Increased Model Complexity: The proposed method adds complexity with multiple modules, which may hinder scalability and efficiency.
2. Generalization to Other Domains: The paper does not sufficiently address how well these methods would perform on other types of imagery or detection tasks.
3. Potential for Overfitting: The reliance on sophisticated techniques like hyperbolic space mapping could increase the risk of overfitting, particularly with limited data.
4. Lack of Interpretability: The paper could improve in explaining how the distillation process affects the student model's decision-making process.
5. Implementation Specifics: More detailed information on implementation and training could enhance reproducibility for the community.
6. Computational Cost: The paper does not discuss the computational requirements, which could be substantial for the proposed method.

**Suitability:**

3

---

### Official Review · Reviewer_wEPt · 2024-05-21

**Rating:** 4
**Confidence:** 2

**Summary:**

The paper introduces a novel KD strategy called IFKD. This strategy revolves around three elements.
（1）	Use the trained MobileSAM to obtain additional knowledge on the input data, thereby enabling the student model to obtain richer knowledge and reducing the student model's dependence on the teacher model.
（2）	Use the Frequency domain reconstruction module to enhance feature representation and reduce background noise.
（3）	The paper believes that the characteristics of teachers and students have many similar details in space, so they use hyperbolic similarity masks to guide student network training
The paper conducted many comparative experiments and ablation experiments to verify the effectiveness of the method.

**Strengths:**

The three key modules are effective and interrelated, enhancing critical information after introducing interference. Then Hyperbolic similarity masks are utilized for fine-grained learning. The Ablation study demonstrates the effectiveness of each module's method, and their combinations also show improvement, providing ample evidence of performance enhancement.

**Limitations:**

The paper presents the Information Fusion with Knowledge Distillation (IFKD) method for fine-grained remote sensing object detection. While the integration of external scene information, frequency domain information, and hyperbolic space information is a novel approach, it would be beneficial to situate this work more clearly within the existing literature.
The paper effectively demonstrates the use of MobileSAM to augment teacher knowledge in the IFKD framework. However, it is not fully clear to what extent the choice of MobileSAM as the auxiliary model influences the distillation performance.

**Suitability:**

2

---

### Official Review · Reviewer_8i8B · 2024-05-21

**Rating:** 4
**Confidence:** 3

**Summary:**

This paper proposes a distillation method for fine-grained object detection in remote sensing images. It includes three key modules: 1) External Disturbance Enhancement (EDE), which uses MobileSAM to enrich teachers’ knowledge and reduce students’ dependency on teachers; 2) Frequency Domain Reconstruction (FDR) to amplify key feature representations and reduce background noise interference by resampling low-frequency information; 3) Hyperbolic Similarity Mask (HSM) to increase intra-class differences, guiding students in analyzing and utilizing teachers’ knowledge, and leveraging the exponential capabilities of hyperbolic space for performance improvement.

**Strengths:**

1. Novelty: This paper proposed a Hyperbolic Similarity Mask (HSM), which utilizes the properties of hyperbolic space to increase intra-class differences, guiding the student model in better analyzing and utilizing the teacher’s knowledge.
2. Theoretical Approach and Technical Correctness: The paper’s theoretical approach is sound and well-founded, such as the existing MobileSAM, frequency domain analysis and hyperbolic geometry.
3. The abstract is clear and well-structured enough to express the key points. Additionally, each module of the IFKD framework is described in detail, with clear explanations of their roles and contributions.

**Limitations:**

1. Performance Metrics: While the paper uses AP50 as a performance metric, additional metrics such as precision, recall, F1-score, and computational efficiency would provide a more rounded assessment of the method's performance.
2. More visual results and comparison methods should be increased in Section 4.6 to sufficiently highlight the “fine-grained” remote sensing object detection.
3. Computational Efficiency: Provide a detailed analysis of the computational overhead introduced by the proposed method. Discuss its feasibility for real-time applications and potential optimizations.
4. There are some spelling and grammar errors I noticed in the paper, such as lines 366, 445.

**Suitability:**

2

---

### Official Review · Reviewer_2ACK · 2024-05-23

**Rating:** 4
**Confidence:** 3

**Summary:**

This paper proposes a designed Information Fusion with Knowledge Distillation methods for fine-grained remote sensing object detection. By integrating information from external images, frequency domain, and hyperbolic space, the challenges of heavy background noise and intra-class similarity are solved. The paper is well-written and organized.

**Strengths:**

1. The method is well-designed for a specific domain.
2. The methods illustrations are clear.
3. The experiment and the ablation study are somehow adequate.

**Limitations:**

1. The reasons why authors choose these three additional information need more illustrations.
2. Fig. 1 (b) cannot explain the complex background in remote sensing data. It is not direct.
3. Authors claim for both accuracy and efficiency, the knowledge distillation is used. However, the efficiency performance is not mentioned in the experiment results.

**Suitability:**

2

---

### Meta-Review · Area_Chair_d8nw · 2024-06-30

**Recommendation:** Accept (Poster)
**Confidence:** 4

**Metareview:**

The draft received one weak accept, two borderline accept and one weak reject. The novelty and experimental validation are in general appreciated by the reviewers, while concerns were raised on some technique details, complexity, etc. The authors' rebuttal addressed some concerns but overall did not affect much the final rating. After carefully checking all reviews and authors' feedback, we agree with the reviewers on the technique contributions and follow the majority voting to recommend the work to ACM MM.